# *Lycium barbarum* Glycopeptide Inhibits Colorectal Cancer Cell Proliferation via Activating p53/p21 Pathway and Inducing Cellular Senescence

**DOI:** 10.3390/ijms26157091

**Published:** 2025-07-23

**Authors:** Meng Yuan, Da Wo, Yuhang Gong, Ming Lin, En Ma, Weidong Zhu, Dan-ni Ren

**Affiliations:** Academy of Integrative Medicine, College of Integrative Medicine, Fujian Key Laboratory of Integrative Medicine on Geriatric, Fujian University of Traditional Chinese Medicine, Fuzhou 350122, China; marenyuan2024@gmail.com (M.Y.); dwo_work@126.com (D.W.); yuhang_galal@163.com (Y.G.); linming1479@163.com (M.L.); 18217497624@163.com (E.M.); wzhu@tongji.edu.cn (W.Z.)

**Keywords:** *Lycium barbarum* glycopeptide, colorectal cancer, cell cycle arrest, p53/p21 signaling pathway

## Abstract

Colorectal cancer (CRC) is one of the leading causes of cancer-related deaths worldwide. Its sustained proliferative signaling poses a major challenge for effective therapeutic intervention. Since CRC originates from aberrantly proliferating crypt cells, limiting proliferation or inducing senescence may offer a promising treatment approach. *Lycium barbarum* glycopeptide (LbGP), a traditional Chinese medicine component, is known for its immunomodulatory and other beneficial effects. This study aims to examine the anti-tumor effects of LbGP in CRC as well as its underlying mechanisms of action. We used CT26 CRC cells to investigate the effects of LbGP on tumor proliferation both in vitro and in an allograft mouse model. LbGP treatment significantly inhibited CT26 cell proliferation in vitro and suppressed tumor growth in CT26-implanted mice. Furthermore, LbGP treatment significantly upregulated p53/p21 levels both in vitro and in vivo, leading to CT26 cell cycle arrest in the S phase and the induction of tumor cell senescence. These findings demonstrate that LbGP effectively induces CRC cell cycle arrest and senescence via the p53/p21 pathway and may serve as a promising candidate for CRC adjuvant therapy.

## 1. Introduction

Colorectal cancer (CRC), a malignant tumor of the colon and rectum, is the third most prevalent cancer worldwide and the second most common cause of cancer-related deaths [1]. Although the five-year survival rate for early localized CRC can be as high as 90%, metastatic CRC has a markedly lower survival rate of only ~10% [2]. The period during which CRC progresses from adenoma to adenocarcinoma is a crucial timeframe for early intervention [3,4]. However, other than standard chemotherapy and radiotherapy procedures, there is a lack of effective drugs that can effectively limit CRC progression. Hence, the development of new drugs that target CRC is crucial to improve patient survival and treatment outcomes.

CRC originates from abnormal cell proliferation that is closely associated with dysregulation in the cell cycle [5]. A major player involved in CRC progression is the overexpression of cyclin-dependent kinases (CDKs) [6]. CDKs are mainly regulated by p53, the master regulator of cell cycle regulation by controlling the CDKs inhibitor p21, which directly inhibits CDKs activity during cellular stress, thereby preventing tumorigenesis [7,8,9]. Sustained activation of the p53/p21 signaling pathway leads to cell cycle arrest and induction of cellular senescence, hence targeting the p53/p21 pathway is a promising aspect in the therapeutic treatment of various cancers including CRC [10,11]. In this study, we utilized the murine CT26 CRC cell line that is noted for its strong subcutaneous tumorigenicity and the presence of the wild-type *TP53* gene, and has been extensively used in CRC drug research and development [12,13].

*Lycium barbarum* L. is a well-known herb that has been used in traditional Chinese medicine in boosting immune response as well as immune modulation in diseases such as cancer [14]. Its major bioactive component, *Lycium barbarum* polysaccharide (LBP), has been shown to exhibit marked anti-tumor activity [15]. Notably, LBP induces G0/G1 cell cycle arrest in SW480 and Caco-2 CRC cells [16]. Interestingly, its derivative, *Lycium barbarum* glycopeptide (LbGP), has been shown to exert comparable effects at much lower doses [17]. LbGP exhibits diverse pharmacological activities, including the ability to inhibit the proliferation and growth of MCF-7 breast cancer cells [18], inhibit glioblastoma growth in nude mice [19], promote phagocytic activity of RAW 264.7 macrophages, and modulate gut microbiota composition in murine models of colitis [20,21]. However, the effects of LbGP on CRC and its underlying mechanism of action remain unexplored.

In this study, we examined the anti-tumor effects of LbGP and further hypothesize that LbGP can inhibit the proliferation of CT26 CRC cells and promote cancer cell senescence by activating the p53/p21 pathway.

## 2. Results

### 2.1. LbGP Inhibits CT26 Cell Proliferation

We first examined the potential effects of LbGP on the growth and proliferation of CT26 CRC cells. Normally, CT26 cells are spindle shaped, with complete morphology; however, treatment with LbGP for 72 h resulted in marked changes in cell morphology, including decreased cell density and formation of elongated cells, and this phenomenon was more obvious at higher concentrations of LbGP (Figure 1A). Although the total number of cells were visibly decreased at these higher concentrations of LbGP, Trypan blue staining showed that the percentage of viable cells actually remained stable (Figure 1B, left panel), while the total number of CT26 viable cells was significantly reduced in a dose-dependent manner (Figure 1B, right panel). In addition, the Cell Counting Kit-8 (CCK-8) assay further demonstrated that LbGP significantly inhibited the proliferation of CT26 cells in a dose-dependent manner (Figure 1C). In contrast, non-cancerous cell lines exhibited no marked morphological changes or decreases in cell viability (Appendix A). These results suggest that LbGP treatment significantly reduced the number of CT26 viable cells via suppressing cancer cell proliferation rather than cytotoxicity-induced cell death.

### 2.2. LbGP Reduces the Clonogenic Capacity of CT26 Cells

To further validate the anti-tumor effect of LbGP, we performed a colony formation assay in CT26 cells under anchored and non-anchored conditions for 14 days. Under normal conditions, individual CT26 cells formed numerous colonies under both conditions, whereas cells treated with LbGP had significantly fewer colonies as well as smaller sized clonospheres (Figure 2A,B). Further, CT26 cells cultured in monolayer and treated with LbGP also showed a significant decrease in colony formation rate compared to Phosphate-buffered saline (PBS)-treated cells, and in a dose-dependent manner, with a 77% reduction observed at a concentration of 500 μg/mL (Figure 2A, right panel). These results demonstrate that LbGP can markedly reduce the number of malignantly transformed cells capable of growing in an anchorage-independent manner (Figure 2B), which provides a more accurate prediction of their tumor-forming ability in vivo. Hence, LbGP can inhibit the anchorage-independent growth of CT26 cells, thereby reducing the clonogenic ability of tumor cells.

### 2.3. LbGP Arrests Cell Cycle of CT26 via Upregulating p53/p21

Tumor cell survival or death often depends on the balance between the apoptosis and cell cycle arrest signaling pathways that are regulated by the master regulator p53. As we showed above, there were no significant differences in CT26 cell viability following LbGP treatment, suggesting that LbGP primarily inhibits CT26 cell proliferation via cell cycle arrest. Hence, to determine the molecular mechanisms underlying these changes, we measured the levels of p53 protein and its downstream CDKs inhibitor p21. Western blot analysis revealed that treatment with LbGP significantly increased both the expressions of p53 and p21 in CT26 cells, and in a dose-dependent manner that peaked at approximately 250 μg/mL (Figure 3A). In addition, a p53 inhibitor, Pifithrin-α, completely prevented the ability of LbGP in inhibiting CT26 cell proliferation (Appendix A), further supporting our findings that LbGP’s anti-tumor effects were via upregulating the p53 signaling pathway.

Next, we performed flow cytometry analysis to examine the direct effect of LbGP on the CT26 cell cycle. Notably, the proportion of CT26 cells in the S phase was significantly increased following LbGP treatment. Furthermore, the proportion of cells that entered G2 phase was significantly reduced from 20.25% in control CT26 cells to 16.5%, 11.45%, and 5.93% in the 125, 250, and 500 μg/mL LbGP treatment groups, respectively (Figure 3B). The effect of LbGP on G0/G1 phase was not apparent. Taken together, these findings demonstrate that LbGP intervention induces CT26 cell cycle arrest in the S phase, thereby inhibiting cancer cell proliferation.

### 2.4. LbGP Promotes Cellular Senescence In Vitro

CT26 cells that were cultured for 10 days and treated with LbGP displayed typical cell morphology that is typical of senescent cells, characterized by multiple nuclei and a broad, flattened cell appearance (Figure 4A), and this phenotype was more obvious at higher concentrations of LbGP. Next, we examined the effect of LbGP on CT26 cellular senescence via senescence-associated β-galactosidase (SA-β-gal) staining. CT26 cells treated with LbGP had a significantly greater number of SA-β-gal positively stained cells, and in a concentration-dependent manner (Figure 4B). Taken together, these observations are consistent with the characteristics of senescent cells, suggesting that LbGP can induce cell cycle arrest leading to the senescence of CT26 CRC cells.

### 2.5. LbGP Suppresses Tumor Growth in CT26 Tumor-Bearing Mice

We next examined the anti-tumor effects of LbGP on CRC in vivo using a mouse allograft model. Mice implanted with CT26 CRC cells resulted in a linear increase in tumor volumes over a period of 21 days, which were significantly inhibited in mice treated with LbGP (both low- and high-dose groups) from day 14 and became more obvious by day 21 post treatment (Figure 5A). Notably, LbGP did not result in significant changes in tumor-free body weight (Figure 5B) or major organ weights—heart, liver, spleen, lungs, and kidneys (Appendix A), which suggests that LbGP treatment exhibited no apparent toxicity. Moreover, LbGP-treated mice had significantly reduced tumor sizes as observed following euthanasia (Figure 5C), as well as obvious decreases in measured tumor weight that was significant in the high-dose (100 mg/kg) group (Figure 5D). These results demonstrate that LbGP can suppress CRC growth in tumor-bearing mice and in a dose-dependent manner.

### 2.6. LbGP Inhibits Cell Proliferation in Tumor Tissues via Upregulating p53/p21

We further examined the underlying mechanism of LbGP in preventing CRC cell proliferation in vivo. Western blot analysis on tumor tissues of CT26-implanted mice showed that LbGP treatment significantly increased the level of γ-H2AX, a marker of DNA damage-associated senescence (Appendix A), as well as the levels of p53/p21 expressions, with more pronounced upregulation in the high-dose group (Figure 6A). This demonstrates that LbGP promotes tumor senescence and inhibits CRC progression via upregulation of the p53/p21 signaling pathway.

Next, we examined the level of Ki-67 expression, a key marker of cell proliferation, in the tumor tissues of mice implanted with CT26 cells. Immunohistochemical staining showed that tumor tissues in CT26-implanted mice had high levels of active cell proliferation at 3 weeks post implantation, indicated by the obvious expression of Ki-67 staining (Figure 6B). However, LbGP-treated mice showed significant reductions in the number of Ki-67 positive cells in the tumor tissues, and in a dose-dependent manner (Figure 6B). Taken together, these findings demonstrate that LbGP inhibits CRC cell proliferation in vivo, via upregulation of p53/p21.

## 3. Discussion

Despite recent advancements in CRC prevention and treatment, the incidence and mortality rates of CRC remain among the highest worldwide due to uncontrolled tumor growth and disease progression [22]. Hence, the implementation of comprehensive treatment strategies such as chemotherapy, radiotherapy, or novel therapeutic drugs is essential for preventing CRC disease progression and enhancing patient survival [2]. One of the potential novel therapeutic drugs may be *Lycium barbarum* L. that has long been used in traditional Chinese medicine as a complementary and alternative therapeutic agent for boosting immune response. Recent studies have also shown that LbGP exhibits positive effects in modulating immune response in mice [15,23], as well as the ability to inhibit glioblastoma development, while exhibiting no adverse side effects [19]. However, the direct effect of LbGP in inhibiting CRC progression, in particular the underlying mechanism involved, has not been previously reported.

Natural polysaccharides and glycopeptides display diverse biological activities that are influenced by their sugar composition and glycosidic linkages [24]. Recent studies have shown that the highly branched arabinogalactan structure of LbGP contributes to its biological activity [25], and that its deproteinized derivative LBP-3 can alleviate intestinal mucosal injury by enhancing mucin O-glycan expression [26]. Further studies into LbGP’s structure and functional relationship may help clarify its strong anti-cancer effects.

Abnormally proliferating crypt cells in the colon or rectum are well recognized as a pre-cancerous marker of CRC [5]. Previous studies have shown that cell cycle-related genes play crucial roles in immunotherapy for the treatment of numerous cancers [27,28]. One of the factors associated with the prognosis of CRC is the cell cycle checkpoint score, where patients exhibiting a low score are more responsive to first-line CRC chemotherapeutic agents such as 5-Fluorouracil as well as radiotherapy [29]. Therefore, drugs that can regulate the cell cycle may be potential drugs that are beneficial in tumor treatment [27]. In this study, we observed that LbGP arrested CT26 cells in the S phase of the cell cycle and prevented the progression to the G2 phase. Cell cycle arrest serves as a self-protective mechanism for accurate cell replication to occur, while senescence represents the eventual cell fate when cell damage becomes irreparable [6]. Hence, our results showed that LbGP induced cell cycle arrest in CRC as well as the hallmark features of cell senescence, thereby demonstrating the robust ability of LbGP in preventing the proliferation and growth of CRC.

Additionally, one of the main regulators of the cell cycle is p53, which acts as a potent tumor suppressor protein, and hence is a main target of novel therapeutic drugs in the treatment of various cancers [30]. Further, p53 downregulates the expression of Ki-67, a key marker of cell proliferation, via a p21-dependent manner [31]. Thus, activation of p53 expression can induce cell cycle arrest and senescence of cancer cells, thereby promoting tumor clearance [7]. In contrast, the knockdown of p53 has been shown to accelerate CT26 tumor growth [32], supporting its importance in tumor suppression. This aligns with our findings that LbGP treatment significantly upregulated levels of p53/p21 proteins, thereby causing tumor cell cycle arrest in S phase and CT26 cell senescence both in vitro as well as in vivo using CRC-bearing mice. Future investigations utilizing LbGP should be investigated in other cancer models in order to further elucidate its robust anti-tumor effects.

Of note, the p53/p21 and p16/retinoblastoma signaling pathways play both synergistic and distinct regulatory roles in cellular senescence [33]. A previous study showed that p21^high^ and p16^high^ cells represent distinct subpopulations of senescent cells [34]. In our present study, p16^INK4a^ was undetectable in tumor tissues of CT26 tumor-implanted mice, which can be explained by the fact that the *CDKN2A* gene that encodes p16^INK4a^ is known to be deleted in a homozygous manner in CT26 cells [13]. These studies support our findings that the cancer senescence-inducing ability of LbGP in CT26 colorectal cancer was primarily mediated via the p53/p21 signaling pathway. Taken together, our current study revealed the robust ability of LbGP in inhibiting CRC cell proliferation and promoting senescence, providing new evidence for the potential use of LbGP as a novel and effective drug for the treatment of CRC.

## 4. Materials and Methods

### 4.1. Materials

*Lycium barbarum* glycopeptide (LbGP, research grade, purity ≥95%) was obtained from Ningxia Tianren Goji Biotechnology Co., Ltd., Zhongwei City, China (lot no. 20180107) and stored in a sealed container at room temperature, shielded from light. LbGP was extracted from dried *Lycium barbarum* berries using alcohol-free extraction, high-speed sedimentation, and membrane separation as previously described [17,21], and outlined in US utility patent US 11110144 B2 [35]. Batch-level documentation is available upon request.

The following antibodies were used: for Western blotting, p53 (1:2000, #2524, Cell Signaling Technology, Danvers, MA, USA), p21 (1:2000, #28248-1-AP, Proteintech, Rosemont, IL, USA), gamma H2AX (1:1000, #ab81299, Abcam, Cambridge, UK), GAPDH (1:5000, #60004-1-Ig, Proteintech), Tubulin (1:5000, #11224-1-AP, Proteintech), HRP-conjugated secondary antibodies (rabbit: 1:5000, #8715; mouse: 1:5000, #6229, Signalway Antibody, Greenbelt, MD, USA); for immunohistochemistry, Ki-67 (1:150, #MAS-14520, Thermo Fisher Scientific, Waltham, MA, USA). The p53 inhibitor Pifithrin-α hydrobromide was purchased from Selleck Chemicals, Houston, TX, USA (S2929).

### 4.2. Animal Experimentation

All experimental protocols involving animals were approved by the Institutional Animal Care and Use Committee (IACUC) of Fujian University of Traditional Chinese Medicine, Fujian, China (Approval No. FJTCM IACUC 2023-097). All procedures were conducted in accordance with the guidelines of the Animal Research: Reporting of In Vivo Experiments (ARRIVE) framework. Healthy male BALB/c mice were obtained from Shanghai SLAC Laboratory Animal Company (Shanghai, China). Animals were housed in a specific pathogen-free (SPF) environment at a temperature of 25 ± 1 °C, relative humidity of 55 ± 5%, and a 12 h/12 h light–dark cycle.

For tumor implantation, murine CT26 CRC cells were washed and resuspended in ice-cold PBS and subsequently each mouse was injected with 100 μL cell suspension (1 × 10^6^ cells per mouse) subcutaneously in the right axillary region. After injection, a total of 24 mice were randomly divided into three groups (*n* = 8 per group): control group (PBS), a low-dose LbGP group (50 mg/kg), and a high-dose LbGP group (100 mg/kg). LbGP was administered once daily by oral gavage, and the control group received an equivalent volume of PBS. Tumor volume was measured every 7 days using digital calipers and calculated using the formula: *V* = (*ab*^2^)/2, where *V* = tumor volume in mm^3^, and ‘*a*’ and ‘*b*’ represent the longest and shortest measured diameters of the tumor, respectively.

### 4.3. Cell Culture

Murine CT26 CRC cell line (RRID: CVCL_0030) was purchased from American Type Culture Collection (ATCC), Manassas, VA, USA. CT26 cells were cultured in RPMI-1640 medium (Thermo Fisher Scientific) supplemented with 10% fetal bovine serum (Gibco, Thermo Fisher Scientific, Waltham, MA, USA) and 1% penicillin-streptomycin (100 μg/mL). Cells were incubated in a humidified atmosphere supplemented with 5% CO_2_ at 37 °C. LbGP powder (180 mg) was dissolved in 40 mL of basal medium, vortexed, and incubated in a 37 °C water bath for 1 h until completely dissolved, then filtered through a 0.22 μm filter membrane to obtain a sterile stock solution of 4000 μg/mL and stored at −20 °C. The stock solution was diluted with culture medium to the desired concentration for experimental use.

### 4.4. CCK-8 Assay

CT26 cells were seeded into 96-well plates at a density of 3000 cells per well and treated with varying concentrations of LbGP (0, 62.5, 125, 250, 500, and 1000 μg/mL). After 72 h, the medium was replaced with 100 μL of fresh medium containing 10% CCK-8 solution (K1018, ApexBio Technology, Houston, TX, USA), a water-soluble tetrazolium salt-based assay for evaluating cell viability [36], and incubated for 30 min at 37 °C in the dark. Subsequently, absorbance was measured at 450 nm (reference: 620 nm), and relative proliferation (% of control) was calculated based on the standard curve.

### 4.5. Trypan Blue Assay

Trypan blue staining was performed as previously described [37]. Briefly, cells were digested in Tris-EDTA, resuspended and diluted to a specific concentration then mixed with an equal volume of 0.4% trypan blue solution (C0040, Solarbio, Beijing, China). Subsequently, a 20 μL sample was loaded onto a hemocytometer and the number of viable cells were immediately counted under a light microscope. For cell counting, 50 cells or fewer per quadrant were counted, corresponding to a final concentration of 5 × 10^5^ cells/mL or lower. All four quadrants in the hemocytometer were assessed and the numbers of viable cells (unstained) and total cells (blue + unstained) were recorded. Cell viability was calculated as the proportion of viable to total cells.

### 4.6. Colony Formation Assay

Colony formation assays were performed as previously described [38]. Briefly, CT26 cells in logarithmic growth were digested and counted, then reseeded at a density of 200 cells per well. Cell medium was replaced every 2–4 days. After 15 days, when visible colonies had formed, cells were washed with PBS, fixed with 4% paraformaldehyde at room temperature for 15 min, and stained with 0.1% crystal violet for 20 min. Cell colonies were subsequently photographed and counted.

### 4.7. Soft Agar Colony Formation Assay

Colony formation assays were performed as previously described [39]. Briefly, agarose was made up to a final concentration with medium containing 10% serum, and 24-well plates were pre-coated with a 0.6% base layer of agar. Upon solidification, plates were subsequently plated with a 0.35% agarose–cell mixture and 50 μL of medium was added every 2–4 days to maintain moisture. After 15 days, colonies were imaged and quantified using an inverted microscope.

### 4.8. Cell Cycle Analysis

Cell cycle analysis was performed using cell cycle and apoptosis analysis kit (C1052, Beyotime, Shanghai, China). Briefly, cells were harvested, washed with pre-cooled PBS, gently resuspended and fixed with pre-cooled 70% ethanol at 20 °C overnight. After centrifugation and ethanol removal, the cells were washed, RNase A was added to eliminate RNA interference, followed by staining with propidium iodide (PI) according to the manufacturer’s instructions, and subsequent cell cycle analysis was examined using flow cytometry.

### 4.9. Senescence-Associated β-Galactosidase Staining

Cell senescence was assessed using SA-β-gal staining kit (C0602, Beyotime, Shanghai, China), utilizing X-gal as the chromogenic substrate, as previously described [40]. Briefly, cells were washed with ice-cold PBS and fixed at room temperature for 15 min. Subsequently, SA-β-gal staining solution was added, and cells were incubated overnight at 37 °C in a humidified and dark environment, prior to visualization and photography of senescent cells.

### 4.10. Western Blot Analysis

Western blotting was performed according to standard protocol. Briefly, cells or tissues were lysed using Nonidet P-40 (NP-40) lysis buffer (N8032, Solarbio), or Radioimmunoprecipitation assay (RIPA) lysis buffer (AR0102, Boster Biological Technology, Wuhan, China), respectively. Protein concentration was quantified using Bicinchoninic acid (BCA) protein assay kit (AR0197, Boster), and an equal volume of proteins were loaded for sodium dodecyl sulfate–polyacrylamide gel electrophoresis (SDS-PAGE) and subsequently transferred onto polyvinylidene difluoride membrane. Following blocking with 5% skim milk at room temperature for 1 h, membranes were incubated overnight with the respective primary antibodies at 4 °C and secondary antibody at room temperature for 1 h. Finally, proteins were visualized under enhanced chemiluminescence. All experiments were independently repeated at least three times with consistent results.

### 4.11. Immunohistochemistry

After systemic perfusion with ice-cold PBS, fresh tumor tissues were excised and fixed in 4% paraformaldehyde at 4 °C for 48 h. The tissues were then washed under running water overnight, followed by stepwise dehydration in ethanol, clearing in xylene for 15 min, before embedding in paraffin and sectioning (6 μm). For subsequent IHC, slides were rehydrated, then subjected to antigen retrieval in sodium citrate solution at 95 °C for 10 min, and permeabilized with 0.25% Triton X-100. IHC kit (KIT-9720) and 3,3′-diaminobenzidine kit (DAB-0031) were purchased from MXB Biotechnologies, Fuzhou, China, and IHC was performed according to standard protocol. Briefly, slides were sequentially incubated with endogenous peroxidase blocker (10 min at RT), blocking solution (1 h at RT), primary antibody (overnight at 4 °C), secondary antibody (1 h at RT), and streptavidin (1 h), followed by DAB staining, counterstaining with hematoxylin, and mounting with aqueous mounting medium. Images were captured at 400× magnification using a Leica smart automatic optical microscope, and six random regions were selected for each sample and analyzed using ImageJ version 1.54p.

### 4.12. Statistical Analysis

Data were analyzed using Graphpad Prism 9 (version 0.1.1), presented as mean ± SD. For comparisons between multiple groups, one-way analysis of variance (ANOVA) was performed, followed by Tukey’s multiple comparison test. *p* < 0.05 was considered as statistically significant.

## 5. Conclusions

These findings demonstrate the robust ability of LbGP in inhibiting CRC proliferation via activation of the p53/p21 pathway and the induction of tumor cell senescence. Further, the ability of LbGP in markedly arresting the cell cycle in CT26 cells further suggests the potential use of LbGP as a potential novel drug in preventing tumor growth and treatment of CRC.

## Figures and Tables

**Figure 1 ijms-26-07091-f001:**
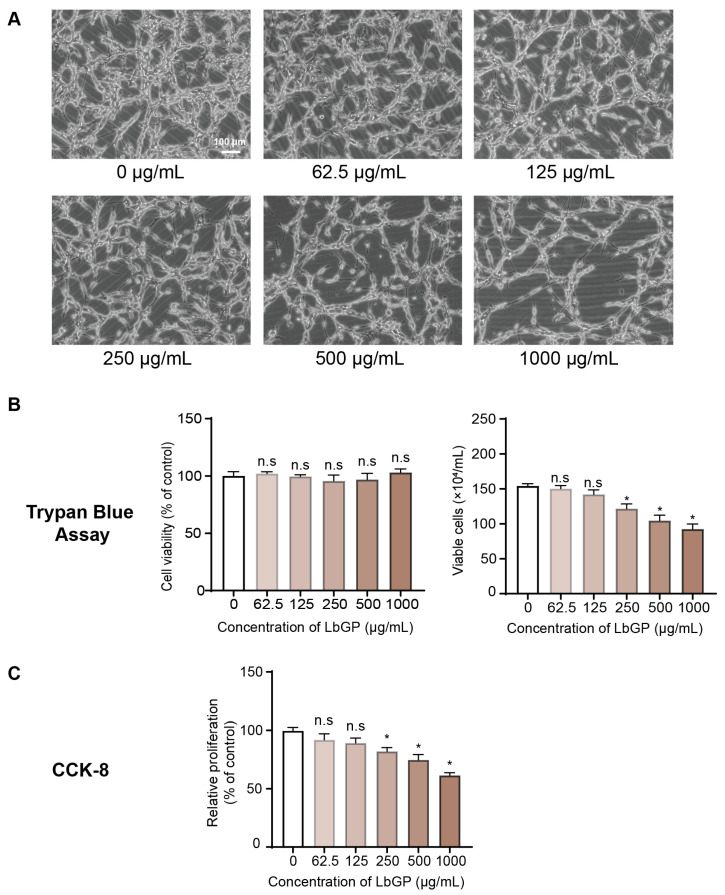
Effect of *Lycium barbarum* glycopeptide (LbGP) on CT26 cell proliferation. (**A**) Representative bright-field images of CT26 cells treated with 0–1000 μg/mL LbGP for 72 h (200×; scale bar = 100 μm, Leica DMIL LED inverted microscope, Wetzlar, Germany). (**B**) Quantification of cell proliferation via Trypan Blue assay. (**Left**): Percentage (%) of CT26 viable cells; (**Right**): Total number of CT26 viable cells. (**C**) Quantification of cell proliferation via Cell Counting Kit-8 (CCK-8) assay. *n* = 3. ** p* < 0.05; n.s., not significant compared to the control (0 μg/mL).

**Figure 2 ijms-26-07091-f002:**
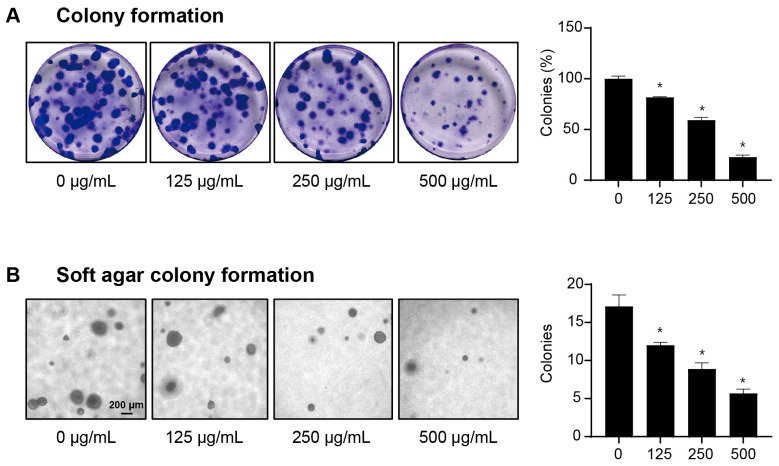
Effect of LbGP on the clonogenic potential of CT26 cells. (**A**) Representative images and quantification of colony formation in plate assays. (**B**) Representative images and quantification of soft agar colony formation (100×; scale bar = 200 μm, Nikon ECLIPSE Ts2 inverted microscope, Tokyo, Japan). *n* = 3. * *p* < 0.05, compared to the control (0 μg/mL).

**Figure 3 ijms-26-07091-f003:**
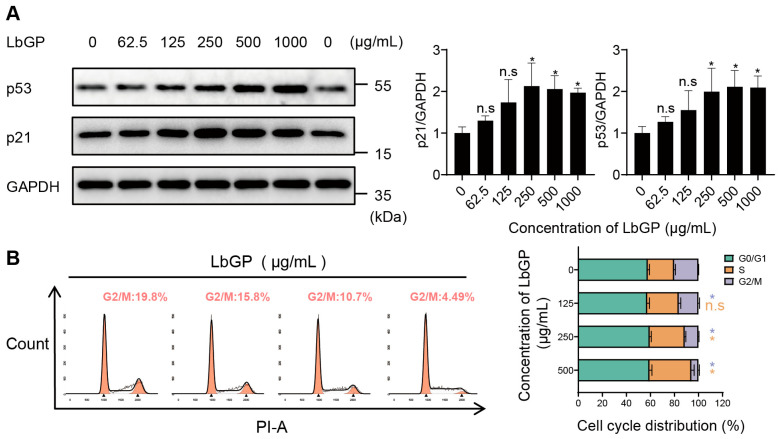
Effects of LbGP on p53/p21 expression and cell cycle distribution in CT26 cells. (**A**) Western blot and band intensity quantification of p53 and p21 expression after 72 h of LbGP treatment. Two 0 μg/mL samples were used as technical replicates for both protein blots. GAPDH served as the loading control. (**B**) Flow cytometry analysis of cell cycle distribution at 72 h, showing an increased proportion of cells in S phase and a decreased proportion in G2/M phase. *n* = 3. * *p* < 0.05; n.s., not significant compared to the control (0 μg/mL).

**Figure 4 ijms-26-07091-f004:**
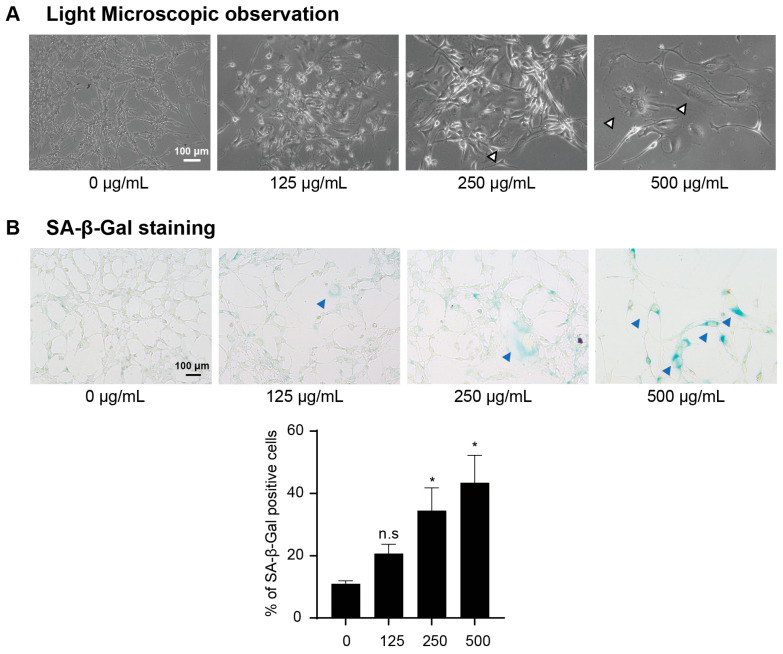
LbGP induces morphological changes and cellular senescence in CT26 cells. (**A**) Representative bright-field images of CT26 cells after 10 days of LbGP treatment (200×; scale bar = 100 μm, Leica DMIL inverted microscope). White arrowheads indicate elongated and flattened cells. (**B**) Senescence-associated β-galactosidase (SA-β-gal) staining (blue) and quantification in control and LbGP-treated cells following 10-day treatment (200×; scale bar = 100 μm, Nikon ECLIPSE Ts2 inverted microscope). Blue arrowheads indicate SA-β-gal-positive cells. *n* = 3. * *p* < 0.05; n.s., not significant compared to the control (0 μg/mL).

**Figure 5 ijms-26-07091-f005:**
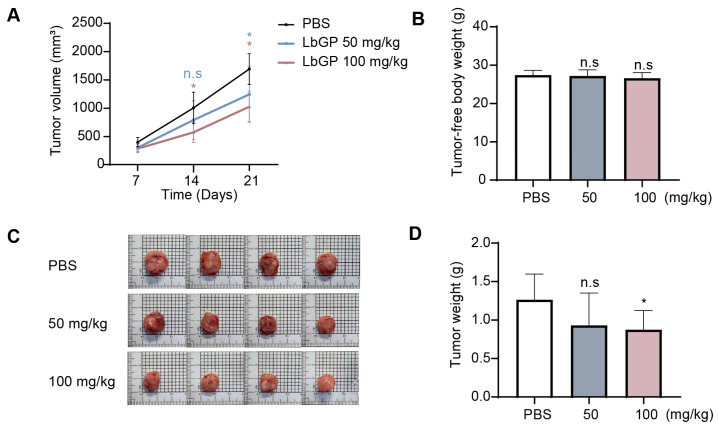
LbGP reduces tumor growth in mouse models of CT26 tumor implantation. (**A**) Tumor volume progression over the 21-day administration of LbGP. (**B**) Tumor-free body weight at day 21 post-tumor implantation. (**C**) Representative images of dissected tumors from each treatment group. (**D**) Tumor weight at day 21 post-tumor implantation. *n* = 8 per group; data represent mean ± SD. * *p* < 0.05; n.s., not significant compared to phosphate-buffered saline (PBS)-treated mice.

**Figure 6 ijms-26-07091-f006:**
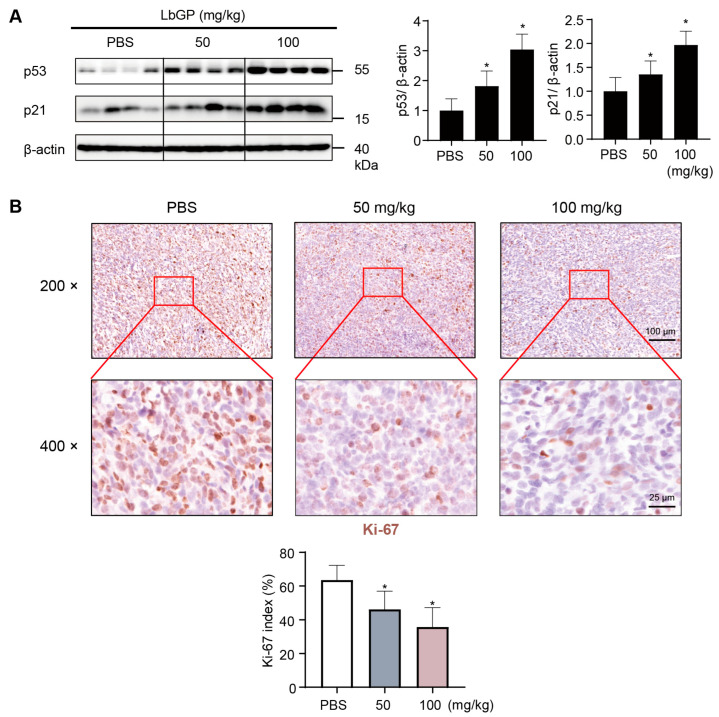
LbGP increases p53/p21 expression and reduces tumor cell proliferation in vivo. (**A**) Western blot quantification of p53 and p21 expressions in tumor tissues at day 21. β-actin served as the loading control. (**B**) Immunohistochemistry (IHC) staining and quantification of Ki-67–positive cells in tumor sections (200× and 400×; scale bars = 100 μm and 25 μm, Leica; antibody 1:150). Brown nuclei represent Ki-67–positive cells. *n* = 8 per group; data represent mean ± SD. * *p* < 0.05; compared to PBS.

## Data Availability

Data are available from the corresponding author on reasonable request and will be made publicly available upon publication in accordance with the journal’s data availability policy.

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
