# Peer review of "Lycium barbarum* Glycopeptide Inhibits Colorectal Cancer Cell Proliferation via Activating p53/p21 Pathway and Inducing Cellular Senescence"

_ijms, 2025, doi:10.3390/ijms26157091_

Round 1

Reviewer 1 Report

Comments and Suggestions for Authors

Through in vitro and in vivo experimental studies, the authors verified that the Lycium barbarum glycopeptide in the traditional Chinese medicine Lycium barbarum glycopeptide can effectively inhibit the proliferation of colorectal cancer cells by activating the P53/P21 signaling pathway, causing the cell cycle to arrest in the S phase and leading to cell senescence.
The topic is relatively novel, has scientific significance, and is also of reference value for the clinical treatment of colorectal cancer.
The manuscript is well designed and written.
But a few suggestions are made:

  1. Only one cell line was used. Generally, at least two or more similar cell lines are used to verify a drug/compound in vitro. In addition to CT26, HT-29 or Caco-2 cells can also be considered.
  2. For in vitro experiments, it is recommended to use a normal cell line, such as CCD-841 cells. Although the authors concluded that the Lycium Barbarum Glycopeptide was non-toxic by comparing the body weight of the treated groups with that of the control group in the in vivo experiment, it may happen that after dissection, it will be found that the internal organs of the mouse, such as the liver, may have developed lesions caused by the high dosage of Lycium Barbarum Glycopeptide.
  3. Errors should be corrected:Line 248: (1 X 106 cells per mouse), Line 253: V= (ab2)/2

Reviewer 2 Report

Comments and Suggestions for Authors

1. The role of p53 independent pathways (such as the p16/RB pathway) has not been ruled out. Although CT26 cells are wild-type p53, the necessity of this pathway needs to be validated through p53 knockout/inhibition experiments.
2. SA - β - gal staining is a primary marker of aging, but other key aging markers (such as gamma H2AX, IL-6/SASP factors, etc.) have not been detected p16INK4a)。 The synergistic effect of p21 and p16 in aging needs to be elucidated.
3. Western blot should provide the original gel with three repeatable results
4. The dilution factor of the antibody used needs to be clearly labeled

Reviewer 3 Report

Comments and Suggestions for Authors

Dear author,

This is a study of significant relevance to the study of cancer; however, I suggest some modifications to enhance your manuscript.

Comments:

Use italics “in vitro” “in vivo”. Please check all your manuscript.

Line 29. I recommend enhancing your introduction, particularly regarding the effects of Lycium Barbarum Glycopeptide on other cancer types.

Dear author, I suggest that Figure 1A be improved to see the effect of elongation mentioned in your text. On the other hand, the figure caption must be described, because you mention the results in paragraphs 76-82, but I don’t know if it is part of the figure caption. I suggest putting in the figure caption description

Example: Figure 1. Effect of LbGP on cell proliferation. A) morphology cellular B)……. C)….

Why didn't you use a non-cancerous cell type to test your concentration?

Figure 2, Improve the figure caption. Paragraphs 98-101 are part of the figure caption or are a description of the results?

Dear author, please improve the figure captions for all the figures

In figure 3A, you have two treatments 0 ug/mL?, why?

Figure 4 A and 4B, please improve the images (make them larger)

In the Western blot assay shown in Figure 6A, the effect on the p53 bands is poorly defined and diffuse.

Improve Figure 6C, to appreciate the effect (visually), because Figure 6D represents the numeric results.

Dear author, why don’t you use a control positive (drug)

In your discussion, you do not relate the structure of the glycopeptide to the observed effect. I suggest including this information to enrich this section. Thanks

Line 230. Lycium barbarum (italics)

Reviewer 4 Report

Comments and Suggestions for Authors

The presented work entitled "Lycium Barbarum Glycopeptide Inhibits Colorectal Cancer Cell Proliferation via Activating p53/p21 Pathway and Inducing  Cellular Senescence“

expanded the knowledge about the mechanism of the Lycium barbarum glycopeptide (LbGP) on the viability and proliferation of the colorectal cancer CT26 cell line. The suppressive effect was also verified on an allograft mouse model. The authors focused on important factors involved in regulation of tumorogenesis : cyclin-dependent kinases (CDKs) which is regulated by p53,  and p21, which directly inhibits CDK. Suppressive effect of glycopeptide on CT26 was demonstrated at higher concentrations.

Overall MS is written in a clear style and with a good level of English. However, additional information should be added to the Results and Methods sections and some issues need to be explained.

Comments:

- Line 67: „Trypan blue staining showed that treatment with LbGP significantly decreased the number of CT26 viable cells in a dose-dependent manner (Figure 1B, right panel), however there were no significant differences in cell viability (Figure 1B, left panel), suggesting that LbGP could inhibit the proliferation of CT26 cells without inducing cell death.“

Question: Why there is discrepancy between results from Trypan blue and CCK-8 viability tests? Given the fact, that results of both tests were calculated as % of untreated control.

Another issue which should be addressed is the use of Trypan blue for quantification of live=viable cells.   The principle of test is that trypan blue is excluded from viable cells and permeates through membrane of dead cells. This is fine for short time of examination but over time (from 5 min onwards) trypan blue permeates cells membrane and cells die gradually what can decrease accuracy of results. This observation was reported by several studies and is by my personal experience using various primary cells.

The information should be added to section Methods how many (in total ) cells from each sample were counted using a hemocytometer for further calculation of % viability and statistical analysis.

Results:

-The number of replicates for statistical analyses is very small, 3 or 4, which is the minimum. Have the in vitro experiments been repeated with a different batch of product? This would increase statistical significance.

-Line 246: Add information about total number of mice and number in groups in section Methods. This is indicated only in Legend to Fig. 6.

-Fig.1: there is not scale bar showing magnification in A- plate of cell images. If possible, add scale bar into images and write size in the legend. Add what type of equipment was used to obtain photographs?

-Line 90: significant decrease in colony formation rate with a 77% reduction observed at a concentration of 500 μg/mL (Figure 2A, right panel).  This quite high concentration. Is it possible to achieve such concentration in vivo in colon tissue?

-Fig 4: Images of cells are quite small and it is difficult for readers to clearly see differences. I suggest to enlarge the size.

Methods.

The topic of MS was to study effect of LbGP, but there is very little information about this product and its quality and purity.  On line 228 they write „The LbGP used in this study was obtained from Ningxia Tianren Goji Biotechnology and stored in a dry container at room temperature (RT), shielded from light.“

Information should be provided in MS whether this product is officially registered and whether concentration and quality of each batch is certified.  Can you write catalogue number? The authors refer to following references (28 and 29) done probably with different  batch.  What was purity of used LbGP?  Sufficient information must be shown in each MS so that study can be repeated by other scientists.

Round 2

Reviewer 2 Report

Comments and Suggestions for Authors

The author has carefully addressed my concerns and supplemented relevant experiments. I believe this article can be accepted in its current form.

Author Response

Thank you very much for your helpful suggestions. We are glad to hear that our revised manuscript is now acceptable.

Reviewer 3 Report

Comments and Suggestions for Authors

Dear author, 

I appreciate your comments being considered for improving your manuscript. I only suggest adding citations to your methodologies.

Regards

Author Response

Thank you very much for your helpful suggestions. We have now added additional citations to our methodologies in our revised manuscript where appropriate (ijms-3738521.doc).